# Actin nano-architecture of phagocytic podosomes

J. Cody Herron [1,2,9], Shiqiong Hu[3,9], Takashi Watanabe[3,8,9], Ana T. Nogueira[3], Bei Liu [3], Megan E. Kern[4], Jesse Aaron [5], Aaron Taylor[5], Michael Pablo [6,7], Teng-Leong Chew [5], Timothy C. Elston [1,2,3] ✉ & Klaus M. Hahn [2,3] ✉

Podosomes are actin-enriched adhesion structures important for multiple cellular processes, including migration, bone remodeling, and phagocytosis. Here, we characterize the structure and organization of phagocytic podosomes using interferometric photoactivated localization microscopy, a super-resolution microscopy technique capable of 15–20 nm resolution, together with structured illumination microscopy and localization-based super-resolution microscopy. Phagocytic podosomes are observed during frustrated phagocytosis, a model in which cells attempt to engulf micropatterned IgG antibodies. For circular patterns, this results in regular arrays of podosomes with well-defined geometry. Using persistent homology, we develop a pipeline for semi-automatic identification and measurement of podosome features. These studies reveal an hourglass shape of the podosome actin core, a protruding knob at the bottom of the core, and two actin networks extending from the core. Additionally, the distributions of paxillin, talin, myosin II, α-actinin, cortactin, and microtubules relative to actin are characterized.

Podosomes are dynamic, actin-enriched structures that form at the plasma membrane to mediate matrix interactions, cell motility, and mechanosensing. They have been observed in macrophages[1,2], bone-degrading osteoclasts[3,4], and immature dendritic cells[5]. Through associations with adhesion proteins[2] they link cellular mechanosensing to signaling and extracellular matrix (ECM) degradation[6,7]. In osteoclasts, podosomes form a sealing zone upon the bone surface to enable bone degradation and resorption[8,9], and podosome dynamics have been shown to play a role in directional migration in macrophages[10], and saltatory migration in osteoclasts[11].

The definition of a podosome, and the differences between podosome-like structures fulfilling different functions, remains unclear. Podosomes and podosome-like structures all appear to consist of a tightly packed actin core surrounded by a looser actin meshwork which includes radial actin fibers that extend away from the core[12]. The structures formed during phagocytosis have been called podosome-like structures[13–15], phagocytic teeth[16] and phagosome-associated podosomes[17]. Phagocytic podosomes recruit proteins like those in other podosomes, but they are shorter-lived, with higher turnover at the phagocytic cup[15]. On closed phagosomes, podosomes were found to be partially resistant to inhibition of the actin nucleator Arp2/3[17], a key component of conventional podosomes. In contrast, the number of phagocytic teeth at the phagocytic cup were significantly reduced upon the inhibition of Arp2/3[16]. Much remains unknown about the nature and role of the podosome-like structures at the phagocytic cup.

[1]Curriculum in Bioinformatics and Computational Biology, University of North Carolina at Chapel Hill, Chapel Hill, NC, USA. [2]Computational Medicine Program, University of North Carolina at Chapel Hill, Chapel Hill, NC, USA. [3]Department of Pharmacology, School of Medicine, University of North Carolina at Chapel Hill, Chapel Hill, NC, USA. [4]Department of Physics and Astronomy, University of North Carolina at Chapel Hill, Chapel Hill, NC, USA. [5]Advanced Imaging Center, Howard Hughes Medical Institute Janelia Research Campus, Ashburn, VA, USA. [6]Department of Chemistry, University of North Carolina at Chapel Hill, Chapel Hill, NC, USA. [7]Program in Molecular and Cellular Biophysics, University of North Carolina at Chapel Hill, Chapel Hill, NC, USA. [8]Present address: Division of Gene Regulation, Cancer Center, Fujita Health University, Toyoake, Aichi, Japan. [9]These authors contributed equally: J. Cody Herron, Shiqiong Hu, Takashi Watanabe. ✉e-mail: telston@med.unc.edu; khahn@med.unc.edu

Cryo-electron tomography showed that the podosome core is composed of a dense network of actin, and that there are radial actin filaments extending from it[18]. Fluorescence super-resolution microscopy has also advanced our understanding of podosome components. Imaging with Airyscan confocal and 3D-structured illumination microscopy (3D-SIM) showed that the podosome core is surrounded by adhesion proteins, including vinculin in a ring at the bottom, and α-actinin covering much of the podosome[7]. In dendritic cells, 2D stochastic optical reconstruction microscopy (STORM) revealed the nanoscale organization of actin and adhesions surrounding podosomes[19]. However, as the present study shows, the resolution of these studies was not sufficient to resolve several salient podosome features.

Interferometric photoactivated localization microscopy (iPALM) can resolve 3D cellular ultrastructure with up to 20 nm lateral (xy) resolution and 15 nm axial (z) resolution[20,21]. Here, we use iPALM to investigate the ultrastructure of phagocytic podosomes in macrophages. We use a frustrated phagocytosis model[22], in which macrophages encounter circular micropatterns of IgG on glass coverslips. The macrophages initiate phagocytosis and attempt to engulf the IgG but are "frustrated" when the IgG remain attached, producing regular arrays of podosomes around the edges of the IgG disk.

In this study, we use the geometric regularity and reproducibility of frustrated phagocytosis to enable automated image analysis, and thereby quantify structural features of podosomes and their associated actin networks. Salient findings include the podosome's hourglass morphology, an actin knob at the podosome base, and actin networks associated with the upper and lower lobes of the hourglass.

## Results

### Distribution of podosomes, IgG, and Fcγ receptor during frustrated phagocytosis

Unless otherwise mentioned, studies were performed using RAW 264.7 macrophages. Actin was visualized either through transfection with Lifeact-HaloTag labeled with Janelia Fluor 549[23], or by staining with phalloidin Alexa Fluor 568. For frustrated phagocytosis (FP) experiments on disks of antibodies, stamps were dimensioned to produce IgG disks of 3–3.5 μm diameter (Methods). Qualitatively similar podosomes were produced by interaction with uniform fibronectin (Fig. 1a), FP on uniform IgG (Fig. 1b), or FP on the IgG disks (Fig. 1c). Primary cells also produced similar podosomes (Supplementary Movie 1).

For FP experiments on IgG disks, we first examined the relative distribution of actin, IgG, and Fcγ receptors (used by macrophages to bind IgG[22], Methods). FcγRIIA-EGFP was transfected into macrophages and imaged with total internal reflection fluorescence (TIRF)-SIM, together with actin or IgG. As observed previously[22], FcγRIIA co-localized with the IgG disks (Fig. 1c, d). Radial averaging (Fig. 1e) revealed that the mean distance from the center to the edge of the FcγR disk ($1.50 \pm 0.08$ μm, StDev) was less than that of the IgG disk ($1.65 \pm 0.04$ μm, $U = 39.0$, $p = 9e-7$) (Fig. 1f). We consistently observed a circle of podosomes beyond the edge of the IgG disk (Fig. 1c–f). The actin peak was $1.81 \pm 0.08$ μm from the center of the disk, further than either FcγR ($U = 0.0$, $p = 8e-9$) or IgG ($U = 0.0$, $p = 8e-9$). Both IgG and FcγR largely tapered away before reaching the phagocytic podosomes (Fig. 1c, d). During FP, podosomes sometimes dissolved and reformed in the same position (Fig. 1g). More rarely, two podosomes merged (Supplementary Movie 2).

### iPALM imaging of phagocytic podosomes during frustrated phagocytosis

Next, we examined the three-dimensional structure of podosomes using iPALM (Fig. 2a, Supplementary Fig. 1, Supplementary Movie 3). Macrophages were plated on coverslips with gold nanoparticles for height calibration (Methods). The cells were allowed to undergo FP on

disks of IgG, then fixed and stained with Phalloidin Alexa Fluor 647. iPALM images were rendered as image stacks from 0 to 550 nm, typically with a Z interval of 10 nm, with the location of the coverslip at $Z = -15$ nm. Examining images and mean actin intensity at different heights clearly revealed distinct structural features (Fig. 2b, c, Supplementary Fig. 1). The actin cores of podosomes were visible as puncta. Examining cross-sections of these cores at different heights showed them to be continuous structures with different diameters and densities at different heights (Fig. 2c, Supplementary Fig. 1a). We saw a relatively small knob of actin protruding from the bottom of podosomes (Fig. 2c, $Z = 20$ nm), below a dense actin network (Fig. 2c, $Z = 70$ nm). At $Z = 200$ nm and 260 nm, the extent of actin outside the core was greatly reduced, with a minimum in the actin intensity distribution at 200 nm (Fig. 2b, c). At $Z = 350$ nm, little actin was seen outside the cores (Fig. 2c).

### Automated identification of phagocytic podosomes and phagocytosis sites using persistent homology

For more accurate quantification, we developed a computational pipeline that could identify podosomes and phagocytosis sites. Using persistent homology[24], a type of topological data analysis, we identified significantly persistent features within an image (Fig. 3a–h, Supplementary Fig. 2, Methods). Simple k-means clustering ($k = 2$) on persistence values was sufficient to automatically find these features (Fig. 3d, e, Supplementary Fig. 2d). This was followed by a refinement step to determine the center of phagocytosis sites and to drop podosomes not associated with a known phagocytosis site (Methods). The pipeline found both podosomes and phagocytosis site centers from actin images alone, required minimal user input with easy to determine initial parameters, and minimized the false discovery rate (Supplementary Fig. 2f, g). When compared to manually identified podosomes from 4 cells, using only the persistent homology step resulted in a false discovery rate of 0.11; including the refinement step resulted in a false discovery rate of 0.01 (Supplementary Fig. 2f, g). The refinement step served several purposes (Methods), but our goal was primarily to minimize false positives.

### Visualizing and quantifying podosome 3D structure

In iPALM images, podosomes and the centers of phagocytosis sites were identified by applying the pipeline to mean Z-projections. Heatmaps of the average actin distribution were generated by analyzing 72 podosomes using two approaches: (1) a line scan perpendicular to the circle of podosomes and oriented towards the site center, and (2) radial averaging (Fig. 4a–d). Using either method, the actin intensity showed an hourglass shape, with an intense upper core, a less intense lower core, and a narrow neck between them (Fig. 4a, b, e). Actin networks extended away from each core (Fig. 4a, b, e). These observations were consistent with our initial analyses of the actin distribution (Fig. 2b, c), and with heatmap visualizations of individual podosomes (Fig. 4f, g and Supplementary Fig. 3a, b).

Using radially averaged heatmaps for individual podosomes, the locations and dimensions of structural features were determined from three quantifications: (1) a contour based on the mean actin intensity (Figs. 4f, 5a), (2) the mean actin intensity for radii between 0 and 100 nm, and thus within the podosome core (Fig. 5a–c), and (3) the mean actin intensity for radii between 400 and 600 nm, and thus away from the podosome core (Fig. 5a–c, Supplementary Fig. 3, Methods).

The podosomes were $330 \pm 20$ nm (StDev) high, measuring from the top to the bottom of the contour. The narrowest part of the neck connecting the two cores was at $190 \pm 10$ nm (Fig. 5d). The upper core was thicker than the lower core (measured using FWHM along the long axis, $90 \pm 30$ nm versus $80 \pm 20$ nm, respectively, $U = 2504$, $p = 7e-5$, Fig. 5e). The width of the upper core was larger than that of the lower core (FWHM along the radial axis: $160 \pm 40$ nm versus $140 \pm 30$ nm, respectively, $U = 2316$, $p = 0.002$, Fig. 5f).

Portions of the lower actin core protruded below the surrounding actin network. The extent of this protrusion was quantified as the distance between the bottom of the podosome and the bottom of the lower actin network (Fig. 5a, Methods), $20 \pm 10$ nm. This ventral protrusion may be important for interactions with the ECM[6,7,25], directly or through other proteins (see discussion).

The lower actin network was thicker than the upper network (FWHM along the long axis, $100 \pm 30$ nm and $70 \pm 20$ nm respectively, $U = 540$, $p = 6e\text{-}12$, Fig. 5e). The most intense section of the upper core was higher than that of the upper actin network ($270 \pm 10$ nm and $260 \pm 20$ nm, respectively, $U = 3536$, $p = 3e\text{-}9$, Fig. 5d), while the most intense section of the lower core was below that of the lower actin network ($70 \pm 20$ nm and $80$ nm $\pm 20$ nm respectively, $U = 2056$, $p = 0.005$, Fig. 5d). Control studies showed that the morphology of the podosomes was not caused by interaction with the edges of the IgG disk (Supplementary Fig. 4).

## 3D volumetric visualization to examine radial fibers

To examine the actin fibers associated with the upper and lower cores, we generated 3D volumetric visualizations of the iPALM data (Methods). Side views of these volumetric renderings showed the ventral protrusion (when viewing 0–80 nm), the lower actin network (80–200 nm), and the upper actin network (200–410 nm) (Fig. 6a). The upper network contained fibers extending radially away from the core, while actin surrounding the lower core was denser and had a more complex arrangement (Fig. 6a, Supplementary Movie 4).

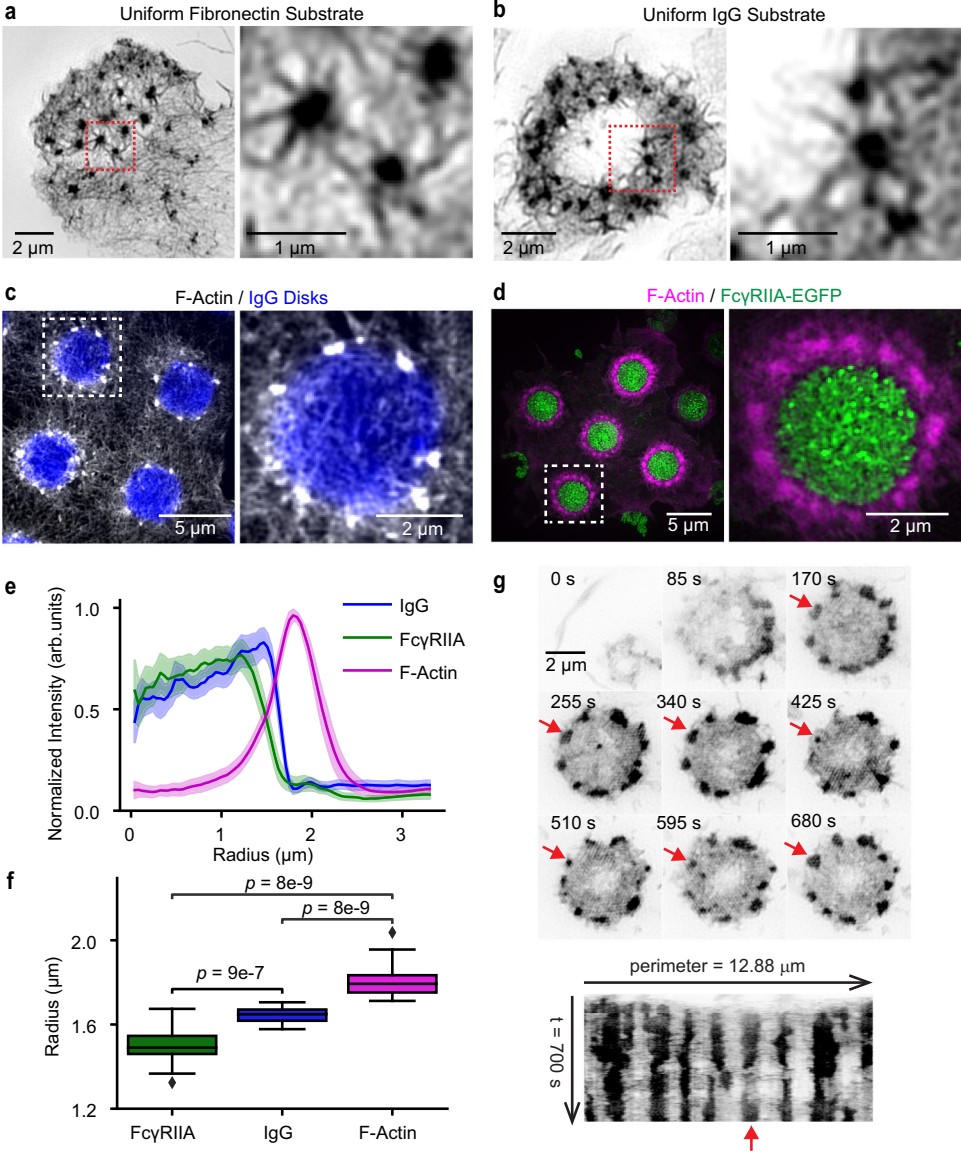

**Fig. 1 | Podosomes on different substrates. a–d** F-actin in RAW 267.4 cells. Enlargements show marked areas (dashed boxes). **a** = uniform fibronectin (phalloidin Alexa 568, SIM). **b** = uniform IgG (phalloidin Alexa 568, SIM). **c** = frustrated phagocytosis on micropatterned IgG disks, showing podosomes organized around the disk. (Lifeact-Halo549, IgG visualized with Alexa647 secondary antibody, TIRF-SIM). **d** = frustrated phagocytosis, showing the FcγRIIA receptor localized over the IgG disk, forming clusters (Lifeact-Halo 549, FcγRIIA-EGFP, TIRF-SIM). **e** Radial profile of proteins from the center of IgG disks (data are presented as mean with 95% CI, $n = 24$ sites from 6 cells). **f** The inflection radius of FcγRIIA-EGFP, IgG, and F-actin measured from individual radial profiles ($n$ as in **e**) showing IgG localization over the IgG disks. Boxes show the median with first and third quartiles. Whiskers show the last datum within 1.5*IQR of the box. *P* values determined using a two-tailed Mann–Whitney *U* test with Bonferroni correction. **g** A circle of podosomes at different time points during frustrated phagocytosis, with kymograph below. Arrow marks podosome that disappears and reappears at the same position (FTractin-EGFP, TIRF-SIM). Source data are provided as a source data file.

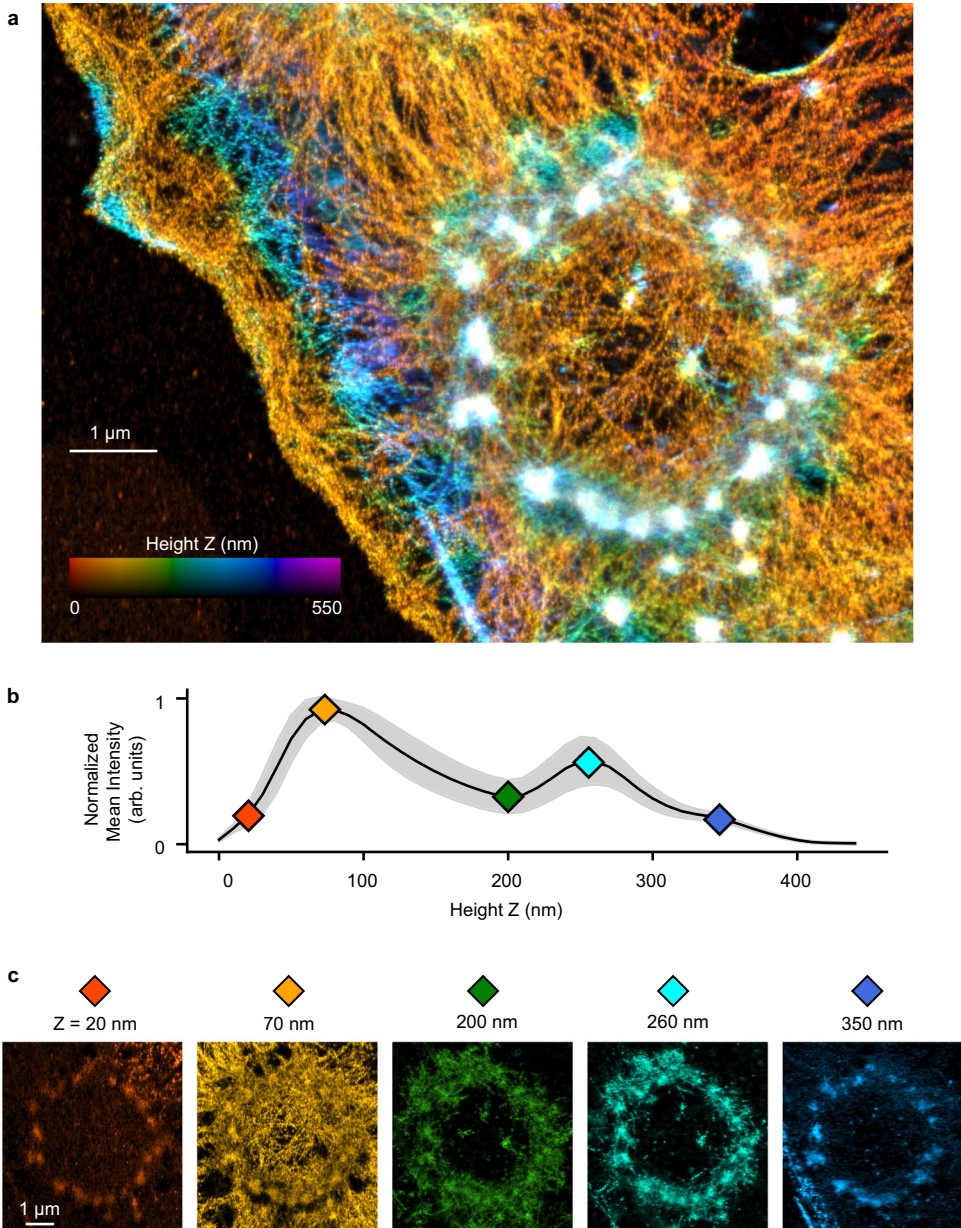

**Fig. 2 | iPALM imaging of actin in frustrated phagocytosis. a** Color coded Z-projection of a frustrated phagocytosis site. Color scale indicates distance of actin from coverslip (Phalloidin Alexa 647). **b** Normalized actin intensity for multiple sites (data are presented as mean with 95% CI, *n* = 7 sites from one cell). **c** Actin images from heights shown by colored diamonds in **b**. Source data are provided as a source data file.

We traced and analyzed individual radial actin fibers using Imaris, a 3D object analysis software (Fig. 6b–d, Supplementary Fig. 5a–c). Qualitative observations indicated that most distinct fibers were in the 200–410 nm height range. The average diameter of these fibers was 30 ± 10 nm (146 fibers from 20 podosomes) (Supplementary Fig. 5c). Additionally, we examined groups consisting of 2–3 podosomes using Z-projections between 200 to 410 nm. Here, we observed overlap between the radial actin fibers extending away from neighboring podosomes (Fig. 6e, Supplementary Fig. 5d, e), although we did not clearly observe a single actin fiber directly connecting two neighboring podosomes.

### Podosome-associated proteins

It has been reported that adhesion proteins form rings around the base of podosomes[26]. We therefore examined the distribution of paxillin and talin using TIRF-SIM (Fig. 7a, b). Qualitatively, each protein appeared more concentrated on the IgG-facing side of podosomes (Fig. 7a, b). Using the computational pipeline described above, we identified podosomes and examined the two proteins using line scans (Fig. 4c), confirming greater concentration on the IgG-facing side (Fig. 7c–e). Whereas talin peaked at the same distance from the podosome center on each side of the podosome (inner: 0.35 ± 0.15 μm and outer: 0.34 ± 0.19 μm, $U$ = 11030, $p$ = 0.52; Fig. 7d), paxillin peaked further away on the inner face (inner: 0.46 ± 0.14 μm, outer: 0.38 ± 0.19 μm, $U$ = 2508, $p$ = 0.004; Fig. 7d). Based on FWHM measurements, peaks for both paxillin and talin were wider on the inside face (for paxillin − inner: 0.25 ± 0.12 μm, outer: 0.18 ± 0.10 μm, $U$ = 2693, $p$ = 8e-5; and for talin − inner: 0.23 ± 0.10 μm, outer: 0.17 ± 0.08 μm, $U$ = 13500, $p$ = 1e-6; Fig. 7e). Consistent with previous observations[27], the paxillin ring had a greater diameter than the talin ring (peak to peak; paxillin = 0.83 ± 0.28 μm, talin = 0.69 ± 0.23 μm, $U$ = 5260, $p$ = 6e-4, Fig. 7f). Qualitative examination of 3D-SIM images also showed that paxillin was more concentrated on the IgG-facing side of the podosome (Supplementary Fig. 6).

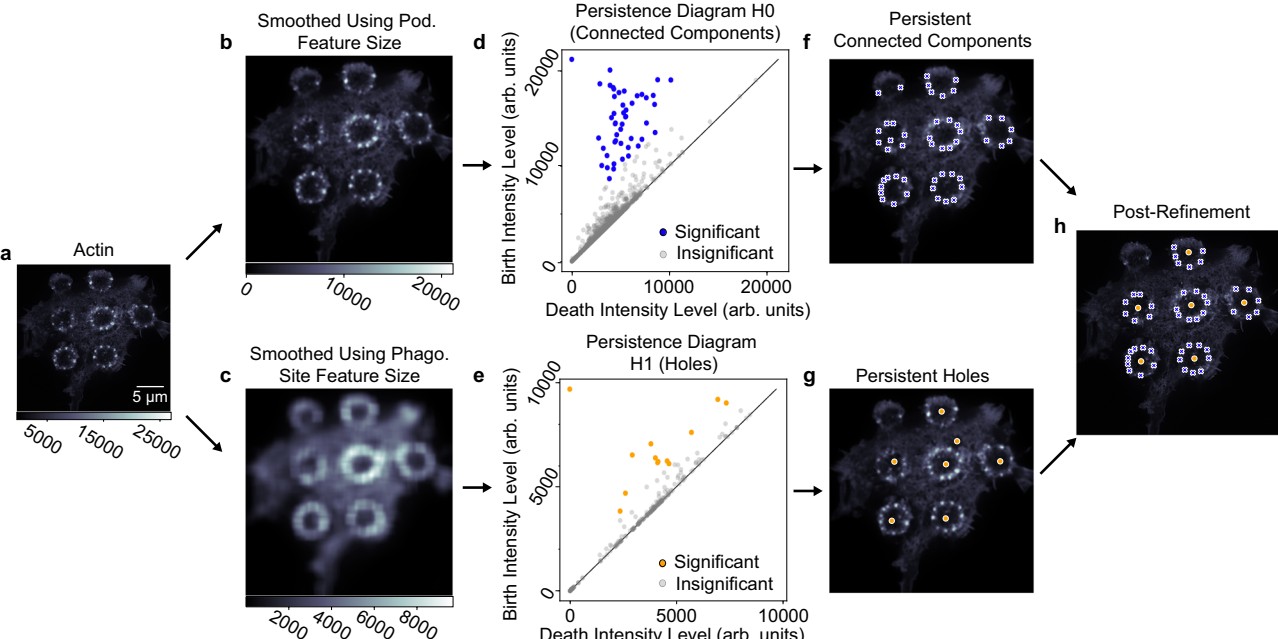

**Fig. 3 | Methods for identifying podosomes and site locations. a** TIRF-SIM imaging of actin in a cell with multiple frustrated phagocytosis sites. Color scale shows pixel intensity (arb. units). **b, c** Smoothed images of the entire cell based on the features of interest (podosomes i.e., pod. or phagocytosis sites i.e., phago.). An insignificant amount of noise is added for uniqueness. Color scales show pixel intensity (arb. units). **d, e** Persistence diagram based on images from **b** and **c**. In **d**, dots (homology-0) represent connected components, and in **e** dots (homology-1) represent holes. Significantly persistent features are colored. Due to pixel uniqueness, the birth intensity of significantly persistent features corresponds to maxima of clustered components and minima of holes. **f, g** Locations of significantly persistent clustered components (**f**) and holes (**g**). **h** Final locations of podosomes and phagocytosis site centers after post-processing to exclude podosomes far away from phagocytosis sites and center phagocytosis sites.

Imaging using 3D-SIM demonstrated that the actin cross-linking protein α-actinin formed a cap-like structure over the top of phagocytic podosomes (Supplementary Fig. 7), as described previously for other podosomes[7].

Cortactin is known to play an important role in podosome formation and function[28,29], so we examined the relative distribution of cortactin and actin. 3D-SIM could not resolve the two-lobed structure, but showed co-localization of cortactin with the actin of the podosome core (Supplementary Fig. 8a, b). Two-color 3D photoactivated localization microscopy (PALM) and STORM did support the existence of a two-lobed structure, with both actin and cortactin in each lobe (Supplementary Fig. 8c–g, Methods).

Because the role of contractility in podosome regulation remains unclear[30], we looked for bipolar filaments of myosin II, important in other forms of cellular contraction. TIRF-SIM was able to resolve the ~300 nm spacing between myosin II heads, so we could use doublets of fluorescent regulatory light chain (RLC-EGFP) to identify myosin II filaments (Fig. 8a)[31]. They formed a ring (Fig. 8a–c) over the IgG disk, with a radius smaller than the ring of podosomes (myosin: $1.06 \pm 0.27 \, \mu m$, actin: $1.83 \pm 0.16 \, \mu m$, $U = 0.0$, $p = 3e\text{-}6$, Fig. 8b, c). 3D-SIM showed that the ring of myosin expanded in radius and approached podosomes as its height increased (Fig. 8d, e, Supplementary Fig. 9, Supplementary Movies 5, 6). The ring did not appear to be continuous; it may have consisted of individual filaments, but this was difficult to discern. TIRF-SIM showed myosin primarily on the inward-facing side (towards the IgG) of podosomes (Fig. 8b), but it was also sometimes observed on the outward-facing side when using 3D-SIM (Supplementary Fig. 9, Supplementary Movies 5, 6).

## Discussion

In this study, we examined the 3D nano-architecture of F-actin in phagocytic podosomes (Fig. 9) using a range of super-resolution fluorescence microscopy techniques. The Z-axis resolution (X, Y: ~20 nm; Z: ~15 nm) of iPALM revealed the structural features that we report. Due to the limited availability of iPALM, we also used TIRF-SIM and 3D-SIM, which have poorer resolution (110 nm lateral and 250 nm axial) but are more accessible and use substantially simpler instrumentation and sample preparation. TIRF-SIM provided higher image contrast and was most useful for live cell imaging due to its acquisition speed. Conventional PALM/STORM filled the gap between iPALM and SIM, with a resolution of 30 nm in the X−Y plane and 50 nm in Z.

Podosomes have been described as actin cones[7,30,32], ranging from 400–700 nm in height[6,7,33]. We instead observed an hourglass shape that, to our knowledge, has not been previously described. This hourglass featured two discrete cores about 200 nm apart, separated by a neck that was typically narrowest closer to the upper core. We measured the height of phagocytic podosomes at $330 \pm 20$ nm, slightly shorter than reported for other podosomes. The differences in shape and size could be due to differences between phagocytic podosomes versus other podosomes, or because the enhanced resolution of iPALM provided a more accurate picture. In our studies here, the decreased resolution of 3D-SIM spread the fluorescence signal in space, leading to height values greater in 3D-SIM than in iPALM.

A ventral protrusion has been postulated to exist on podosomes to facilitate cell-substrate adhesion, interactions with ECM, mechanosensing, and matrix degradation[6,7,25]. We indeed observed an extension of podosome actin below the surrounding actin network. The less organized lower actin network may be cortical actin abutting the podosome. It has been reported that the distance between the bottom of the cortical actin layer and the top of the plasma membrane is less than 20 nm[34], and the plasma membrane itself can have a thickness less than 10 nm[35]. The protruding podosome actin extended $20 \pm 10$ nm below the putative cortical actin. This suggests that that the protruding actin can extend far enough to interact with molecules in the plasma membrane, or potentially poke through it.

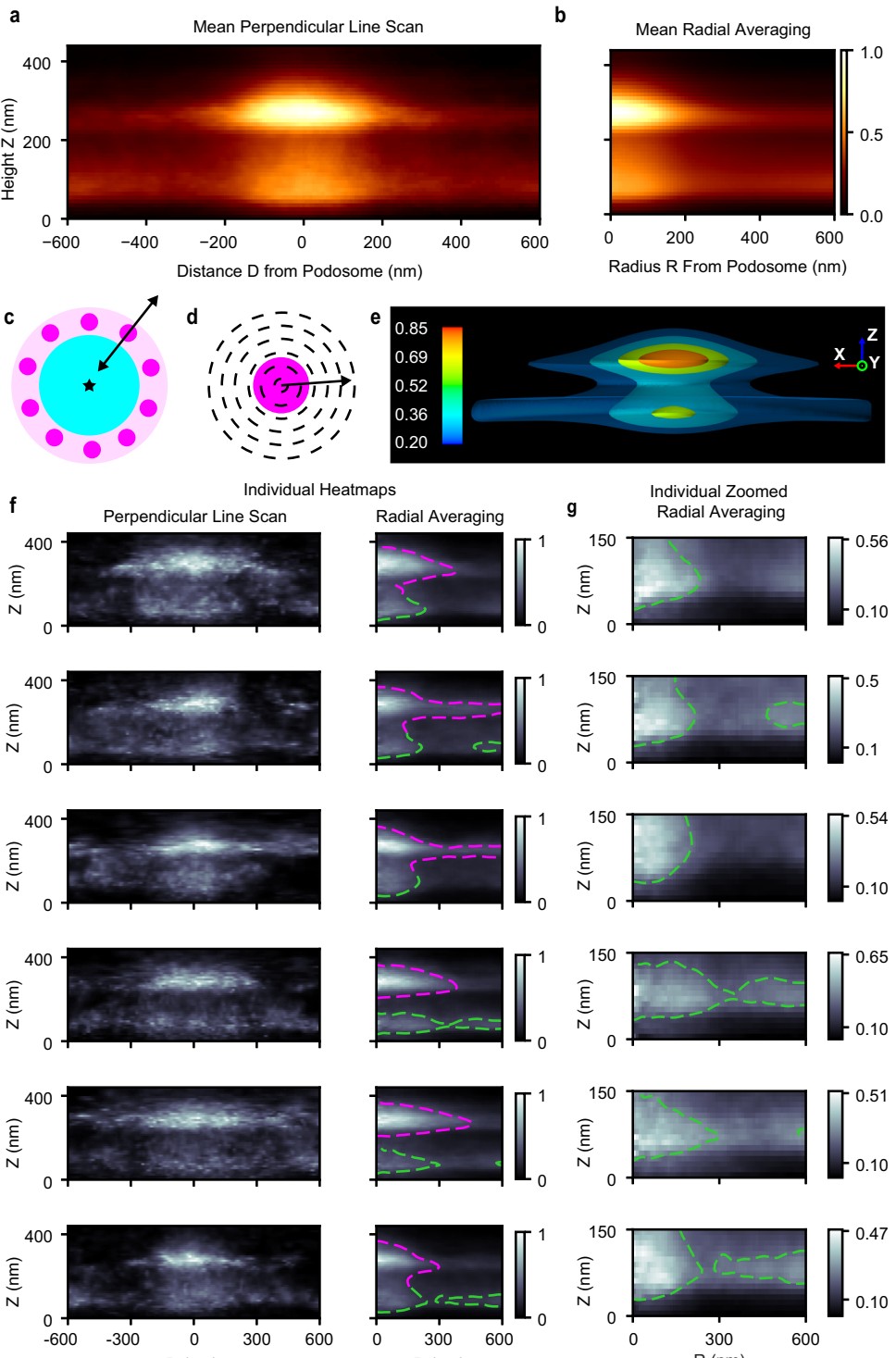

**Fig. 4 | Podosome visualizations reveal 3D nanoarchitecture. a, b** Mean heatmap for $n = 72$ podosomes from one cell using either (**a**) perpendicular line scan (negative D is inward toward the center of the phagocytosis site, positive is outward) or (**b**) radial averaging. Color scale shows normalized mean intensity (arb. units). **c, d** Schematic representations for analyses performed. A perpendicular line scan, as shown in **c** is oriented toward the center of the phagocytosis site, and scans through a podosome perpendicular to the circle of podosomes. Radial averaging, as shown in **d** only depends on the podosome location. **e** Three-dimensional contour rendering of **b** rotated 180 degrees around the Z-axis. Color scale shows normalized mean intensity (arb. units). **f** Perpendicular line scan and radial averaging heatmaps for individual podosomes. Dashed magenta (above 150 nm in Z) and green (below 150 nm in Z) lines are contours based on the mean value within a radius of 350 nm. Color scale shows normalized mean intensity (arb. units). **g** Zoomed view of the podosome protrusions (Z: 0–150 nm), row-wise the same individual podosomes as in **f**. The green contour is the same as in **f**. Color scale shows normalized (as in **f**) mean intensity (arb. units).

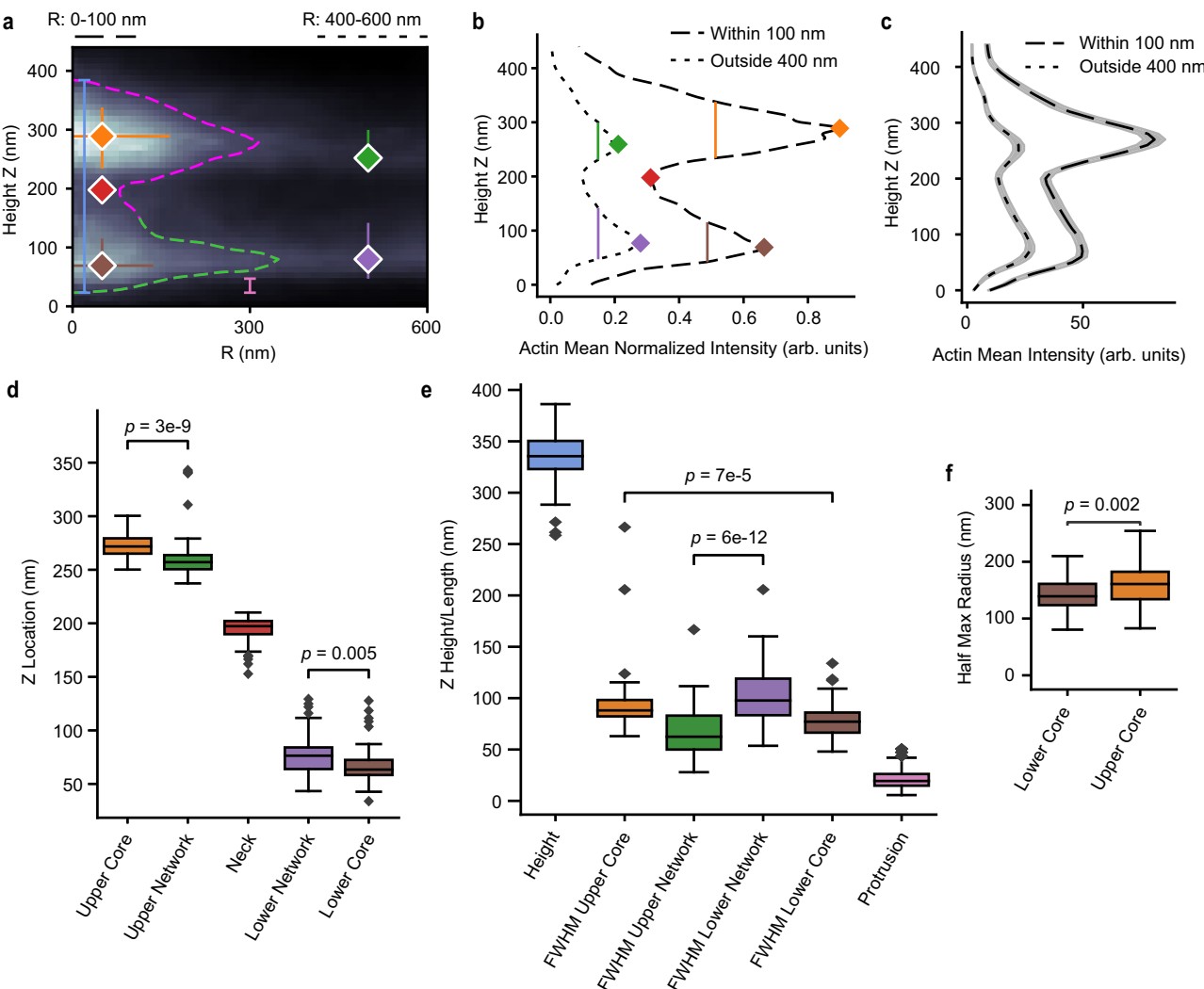

**Fig. 5 | Individual podosome features quantified from iPALM radial averaging heatmaps. a** A radial averaging heatmap for an individual podosome with quantified features. Blue line is podosome height, pink line is protrusion length. Red diamond is the neck location. The location (diamond), Z full-width half-max (FWHM, vertical line), and R half-max radii (horizontal line, only for the cores) are shown for orange = upper core, brown = lower core, green = upper actin network, purple = lower actin network. **b** For the podosome in **a**, mean intensity versus height plot for portions within 100 nm of the center (dashed) and beyond 400 nm (dotted). Peaks, troughs, and FWHM shown correspond to **a**. **c** Mean intensity versus height plots of distributions within 100 nm and outside 400 nm ($n = 72$ podosomes from one cell, data are presented as mean with 95% CI). **d** Z locations of relevant features (diamonds in **a**, **b**). $N = 72$ for the upper core, 61 for the upper network, 61 for the neck, 64 for the lower network, and 48 for the lower core. Boxes show the median with first and third quartiles. Whiskers show the last datum within 1.5*IQR of the box. $P$ values determined using a two-tailed Mann–Whitney $U$ test with Bonferroni correction. **e** Z measurements of relevant features (vertical lines in **a**, **b**). $N = 72$ for the height, 64 for the protrusion, and otherwise as in **d**. Box plots and $P$ values determined as in **d**. **f** Half-max radii for the upper and lower cores (horizontal lines in **a**). N as in **d**. Box plots and $P$ values determined as in **d**. Source data are provided as a source data file.

We observed actin fibers extending radially away from the phagocytic podosomes, as previously described for other types of podosomes[7,18]. The lower portion of the podosome was embedded in a more complex, disorganized actin bed, likely cortical actin abutting the podosome, while the podosome's upper actin core was connected to well-defined individual fibers. We were unable to find a single actin bundle spanning all the way from one podosome to another, as has been proposed[12,32], but overlapping filaments could connect podosomes as well. These connections may be important to transfer signaling molecules, forge a mechanical connection between podosomes, or otherwise coordinate podosome behavior[36]. We saw isolated microtubules within rings of podosomes, moving and unable to escape from the rings (Supplementary Movies 7, 8). This supported the hypothesis that the individual podosomes surrounding IgG spots were connected by a barrier impenetrable to microtubules.

Cortactin co-localization with actin was seen with both 3D-SIM and PALM/STORM. PALM/STORM had the resolution to support the existence of the hourglass morphology and indicated that cortactin was in both lobes. Talin, paxillin, and vinculin, known to form rings around the base of podosomes[7,26,27], showed complex, reproducible distributions relative to the podosome and IgG disk. In normal (i.e., non-frustrated) phagocytosis, these proteins may be more concentrated between podosomes and IgG to play a role in recognition, building phagocytic structures, or the asymmetric application of force. Paxillin formed a wider ring than talin, as previously observed[27], perhaps because it binds to integrins at the plasma membrane while talin bridges paxillin to other adhesions proteins and to actin filaments[20].

Actomyosin contractility has been implicated in podosome formation, stability, and function[7,37], but its role remains unclear[30,38]. Proteins that coordinate actomyosin contractility have been found in a cap over podosomes, including supervillin, α-actinin, and LSP1[30]. Our TIRF-SIM and 3D-SIM data clearly showed a ring of myosin II between the IgG disk and podosomes. 3D-SIM indicated that this ring approached podosomes as its height

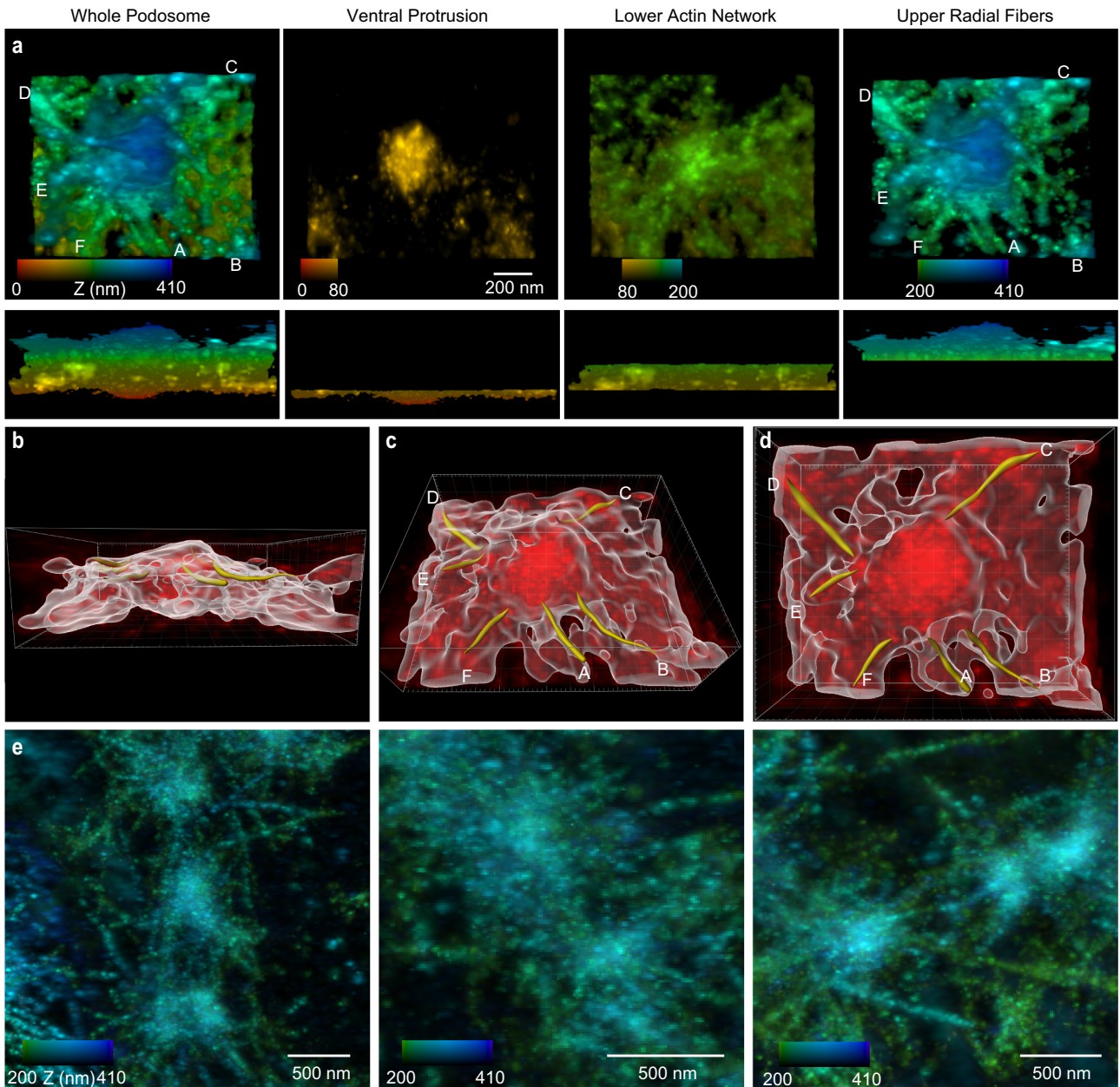

**Fig. 6 | Characterization of radial filaments around individual podosomes.**
**a** Volumetric (3D) visualization of a podosome viewed from the top (top row) and side (bottom row). From left to right: whole image stack, 0 to 410 nm; 0 to 80 nm; 80 to 200 nm; 200 to 410 nm. Radial filaments around the central podosome core are labeled A–F. **b–d** Radial traces for filaments A-F are produced by 3D object analysis (Methods). Traced radial filaments are highlighted in yellow. The dimensions of the bounding box are 1212 x 986 x 410 nm. **e** Three examples from two phagocytosis sites of radial fibers extending away from the podosome core. Z-projected image stacks from iPALM are color-coded from 200 to 410 nm.

increased. In 3D-SIM, but not TIRF-SIM, the myosin II was also sometimes observed on the side of the podosome away from IgG. Thus, our observations were not entirely consistent with the existence of strut-like actin fibers surrounding individual podosomes, as has been postulated[7]. Perhaps these struts were too small or too perturbed to detect. Myosin II may be recruited to the inner face of podosomes, near the IgG disk, to direct protrusive forces[18] for progressing the phagocytic cup.

Finally, we demonstrated semi-automatic identification of podosomes and phagocytosis sites using a persistent homology-based pipeline with simple user inputs. There was no need to manually set a persistence threshold as k-means clustering (k = 2) was sufficient to automatically separate significantly persistent features from short-lived features. Post-processing ensured that we analyzed only

podosomes associated with a well-defined phagocytosis site. These relatively simple steps were able to identify most phagocytic podosomes. The first half of the pipeline, before post-processing, could be easily generalized to other image analysis applications. These techniques enabled quantitation of podosome architecture across many individual podosomes.

The podosome structural features we describe prompt new questions. If there are struts of actin extending from the podosome, where on the two-lobed structure do they attach, how does this affect mechanics, and are signaling molecules arrayed on the two lobes differently? Does the actin network connecting podosomes coordinate podosome behavior, and to what extent is this mediated by signaling molecules or through more rapid mechanical interactions? The podosome shape and fiber attachment points could affect oscillation

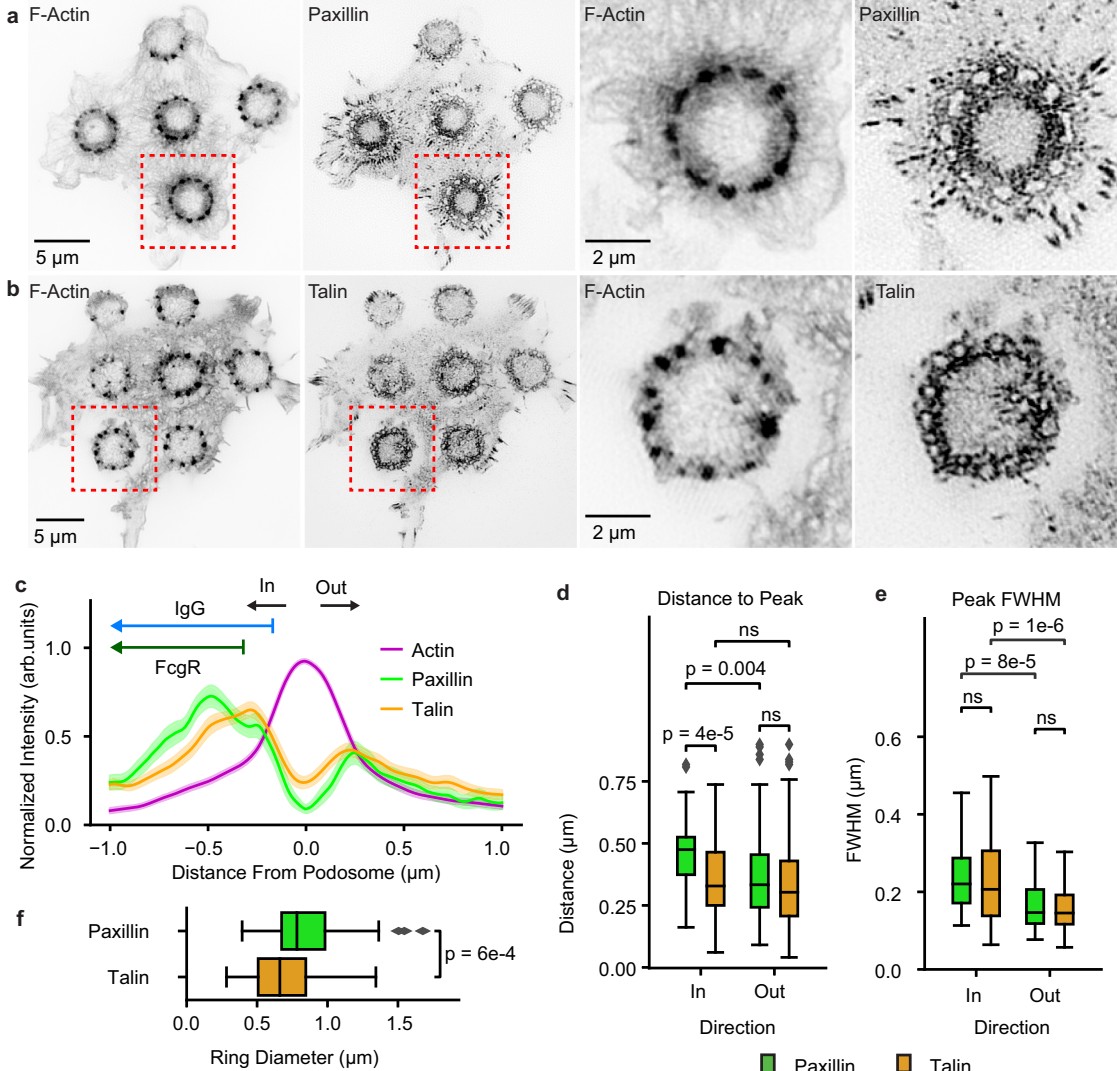

**Fig. 7 | Relative positions of podosome actin core and other proteins. a, b** TIRF-SIM images of F-actin (lifeact-Halo-549), paxillin-EGFP (**a**), and talin- EGFP (**b**) in RAW 264.7 macrophages plated on disks of micropatterned IgG-Alexa 647. **c** Perpendicular line scans for F-actin, paxillin, and talin with podosome center at 0 (Paxillin: $n = 60$ podosomes across 2 cells. Talin: $n = 134$ podosomes across 4 cells). Data are presented as the mean with 95% CI. Arrows for IgG and FcγR show their distribution relative to the actin peak, with the flat end corresponding to the mean inflection points from Fig. 1e, f. **d** Distance from the podosome center to paxillin and talin peaks (n as in **c**). Boxes show the median with first and third quartiles. Whiskers show the last datum within 1.5*IQR of the box. *P* values determined using a two-tailed Mann-Whitney *U* test with Bonferroni correction. **e** Full-width half max of each species peak (*n* as in **c**). Box plots and *P* values determined as in **d**. **f** Distance between the inner (in, toward site center) and outer (out, away from the site) peaks for each species (n as in **c**). Box plots and *P* values determined as in **d**. Source data are provided as a source data file.

frequency[36] or response to different forces. Finally, how does the actin extension at the bottom of the podosome interact directly or indirectly with ECM and membrane components? All these questions will require at least the resolution used here to distinguish the hourglass structure, and ultimately techniques that can report molecular dynamics at this resolution.

## Methods

### Cell isolation and culture

RAW 264.7 macrophages were obtained from the ATCC (TIB-71) and maintained in RPMI 1640 medium supplemented with GlutaMAX (ThermoFisher Scientific, 61870127) and 10% heat-inactivated FBS (HI-FBS, GEMINI Bio, 100–106). The cells were cultured in a 5% $CO_2$ humidified incubator at 37 °C. To detach the cells from Falcon tissue culture dishes (Fisher Scientific, 08-772E), they were treated with Accutase (ThermoFisher Scientific, A1110501) at 37 °C for 5 min before gentle scraping (CytoOne, CC7600-0220). Bone marrow cells were

isolated from the femur of male or female C57BL/6 mice (8–10 weeks age) by using a 5-ml syringe and 20-G needle to pass PBS into the bone marrow and extract the cells. They were cultured in DMEM/F12 medium supplemented with 20% L929 conditioned medium to cause differentiation into macrophages[39,40]. The mice had access to food and water ad libitum and were maintained at constant temperature (22–24 °C), humidity (40–60%), and 12 h light/dark cycles. All animal-related procedures were conducted in accordance with the NIH Guide for the Care and Use of Laboratory Animals and with the approval of the Institutional Animal Care and Use Committee at the University of North Carolina at Chapel Hill.

### Plasmids

pCMV FcγRIIA IRES neo, pEGFP-C1 FTractin-EGFP and EGFP-cortactin were obtained from Addgene (plasmids 14948, 58473 and 26722). FcγRIIA cDNA was cloned into a pEGFP-N vector (Takara Bio USA, CA). EGFP was swapped with mEos3.2[41] in the EGFP-cortactin vector to

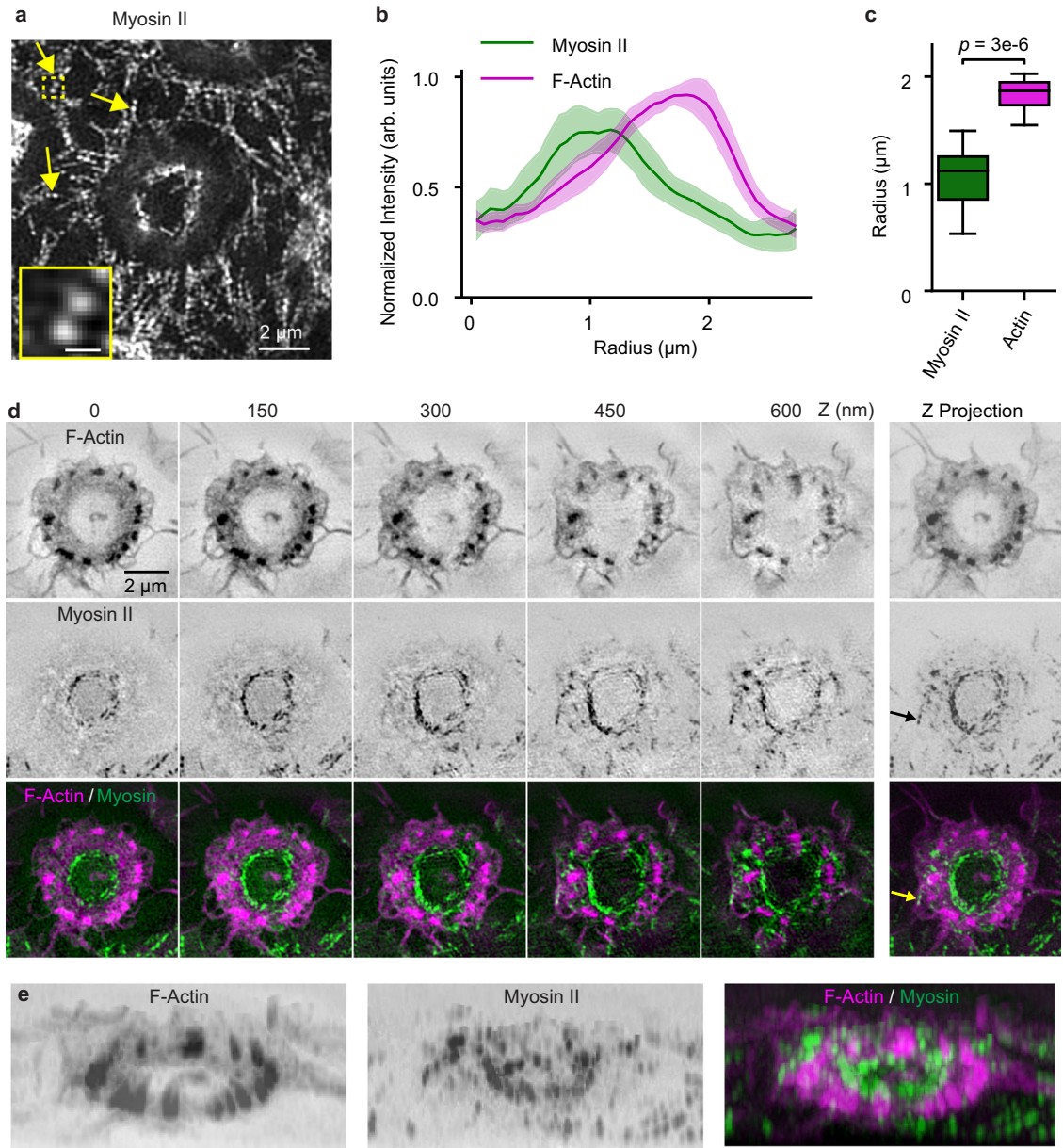

**Fig. 8 | Myosin II filaments in macrophage frustrated phagocytosis.**
**a** Macrophages plated on micropatterned IgG-Alexa 647 disks, with myosin II marked using RLC-EGFP and imaged with TIRF-SIM. Myosin II filaments formed a small ring within the circle of podosomes during macrophage frustrated phagocytosis. Individual myosin II bipolar filaments were visualized as doublets (arrows). The insert, showing a doublet, is a zoom of the dashed box. The scale bar of the insert is 300 nm. **b** Radial distributions of actin and myosin II

($n$ = 15 sites across 7 cells, data are presented as mean with 95% CI). **c** Radial distance to the actin and myosin II peaks. Boxes show the median with first and third quartiles (n as in **b**). Whiskers show the last datum within 1.5*IQR of the box. $P$ values determined using a two-tailed Mann-Whitney $U$ test with Bonferroni correction. **d** Z-stack of actin 3D-SIM images (lifeact-Halo-549), myosin II (RLC-EGFP), and merged. **e** 3D view of actin and myosin II filaments shown in **d**. Source data are provided as a source data file.

---

obtain mEos3.2-cortactin. Lifeact cDNA was fused with Halo tag or mCherry in a pEGFP vector where EGFP was swapped with Halo or mCherry. Lifeact-EGFP was a gift from Michael Sixt (Institute Science and Technology Austria). Paxillin β fused to EGFP was a gift from Ken Jacobson (University of North Carolina-Chapel Hill, USA)[42]. FTractin-tdTomato, myosin regulatory light chain (MRLC)-EGFP, α-actinin-1-mCherry, talin-EGFP, EB3-EGFP, and 3xEMTB-EGFP were described previously[31,43].

**Transfection**
For expressing FcγRIIA-EGFP and Lifeact-mCherry in primary mouse macrophages, FcγRIIA-EGFP or Lifeact-mCherry was cloned into pAdenoX-Tet3G (Takara Bio), and adenovirus was prepared according

to the manufacturer's instructions. Spin infection (300 g for 1 h) was employed to infect primary mouse macrophages with adenovirus[44]. After spinning, doxycycline (final concentration 1.5 μg/ml) was added to the medium and the cells were cultured overnight to induce expression.

For expressing FcγRIIA-EGFP and lifeact-Halo, RAW 264.7 macrophages were transfected with Viromer Red (Lipocalyx, Germany) according to the manufacturer's instructions. Briefly, the cells were seeded into 12-well plates at $2 \times 10^5$ per well one day before transfection. 2 μg of plasmid were mixed with 0.8 μl Viromer Red in Viromer dilution buffer. The cells were incubated with the mixture of plasmid and transfection reagent for 16–24 h before experiments.

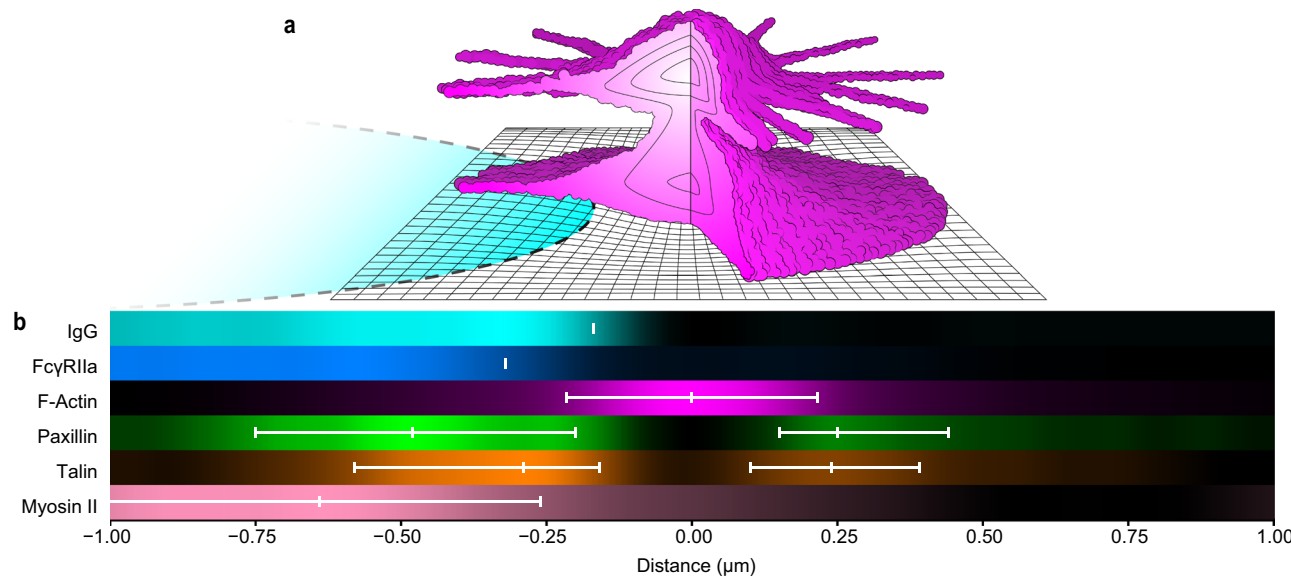

**Fig. 9 | Schematic of podosome constituents drawn to scale. a** Cartoon of actin (magenta) based on iPALM, showing upper radial fibers, lower actin network, and the ventral protrusion. The cross-section shows intensity contours quantified in Fig. 4e. The IgG disk is shown in cyan. Grid unit length = 50 nm. **b** One-dimensional density plots from TIRF-SIM imaging of species centered on the F-actin peak. Plots correspond to the mean values for each species in Fig. 1e (IgG and FcγRIIA), Fig. 7c

(F-Actin, Paxillin, Talin), and Fig. 8b (Myosin II). For IgG and FcγRIIA, white lines correspond to the mean inflection points (Fig. 1f). For paxillin, talin, and myosin, the white lines show the peak and FWHM boundary values. For myosin, the left-hand boundary of the FWHM is beyond the end of the plot but would be at −1.28 µm. PALM/STORM imaging showed cortactin co-localized with actin in both lobes of the podosome (Supplementary Fig. 7).

For expressing FTractin-tdTomato, lifeact-Halo, paxillin-EGFP, talin-EGFP, α-actinin-1-mCherry, RLC-EGFP, EMTB-EGFP, EB3-EGFP, RAW 264.7 cells were electroporated with the Neon Transfection System (ThermoFisher Scientific, MPK5000) following the manufacturer's 10 µl Neon pipette tip protocol. Briefly, $5 \times 10^5$ cells were electroporated with 1–2 µg plasmid in R buffer at 1680 V, 20 ms, with 1 pulse. The cells were transferred into 12-well plates, with each well containing 1 ml of culture medium. For expressing EGFP-cortactin and mEos3.2-cortactin, RAW 264.7 cells were transfected using electroporation with the Neon Transfection System, following the manufacturer's 100 µl Neon pipette tip protocol. $5 \times 10^6$ cells were electroporated with 6 µg EGFP-cortactin or mEos3.2-cortactin in 100 µl R buffer at 1680 V, 20 ms, with 1 pulse. The cells were then distributed into 2 wells of 6-well plates, with each well containing 2 ml of culture medium. After 6–12 h of incubation, the transfected macrophages were ready for experiments. For cells transfected with Lifeact-Halo, 1 µM Janelia Fluor 549 Halo tag dye was added into the cell culture medium for 20 min at 37 C. The cells were then washed and incubated in fresh culture medium for an additional 30 min before microscope imaging. HaloTag dye Janelia Fluor 549 was provided by Dr. Luke D. Lavis (Janelia Research Campus, Howard Hughes Medical Institute).

### Fibronectin or IgG coating
25 mm diameter round glass coverslips #1.5 (Warner, 64-0715) were washed thoroughly with detergent and rinsed with deionized water in an ultrasonic cleaner. Coverslips were further cleaned with a plasma cleaner (Harrick Plasma, PDC-32G) for 2 min immediately before coating with fibronectin. 0.1 mg/ml fibronectin (Sigma, F1141) or 0.1 mg/ml Human IgG (Millipore Sigma, I4506) were coated on the above clean coverslips for 1 h at room temperature.

### Microcontact printing
The IgG patterns on glass coverslips were made using a previously described microcontact printing method[45]. We used a silicon master with an array of 3–3.5 µm diameter holes and center to center spacing of 8 µm, made by photoresist lithography in house.

Polydimethylsiloxane (PDMS) was polymerized on the silicon master using a Sylgard 184 kit (Dow, MI). Polymerized PDMS was cut into ~ 5 × 5 mm square stamps. The stamps were treated with a plasma cleaner (Harrick Plasma, PDC-32G) for 30–60 s. They were then overlaid with 10–15 µl of 1 mg/ml Human IgG (Millipore Sigma, I4506) mixed with either 20 µg/ml Goat anti-Mouse IgG Alexa Fluor 405 (Thermo-Fisher Scientific, A31553) or 20 µg/ml Goat anti-Mouse IgG Alexa Fluor 647 (ThermoFisher Scientific, A21236). Fluid was aspirated from the stamps. They were then placed on the detergent-washed and plasma-cleaned coverslips for 5 min, with an approximately 3 g weight on top. When making IgG patterns on 25 mm diameter iPALM glass coverslips bearing gold nanoparticle fiducial markers[46], a roughly 20 g weight was applied atop the PDMS stamp. IgG-patterned coverslips were prepared fresh on the day of microscope imaging.

### Immunofluorescence staining for fixed macrophage frustrated phagocytosis
Phalloidin Alexa Fluor 488 (dilution 1:500, Invitrogen A12379) and phalloidin Alexa Fluor 568 (dilution 1:500, Invitrogen A12380) were used to stain F-actin for 3D-SIM experiments. Phalloidin Alexa Fluor 647 (Invitrogen; A22287) was used to stain F-actin for STORM experiments (dilution 1:100) and iPALM experiments (dilution 1:25). RAW 264.7 macrophages were plated on IgG-patterned coverslips in Ham's F12 medium (Caisson Labs, UT) supplemented with 2% HI-FBS to enable frustrated phagocytosis for 25 min. For uniform coating experiments, RAW 264.7 macrophages were plated on fibronectin-coated coverslips for 2 h or on IgG-coated coverslips for 1 h. The cells were then fixed with 4% paraformaldehyde at 37 °C for 10–15 min and permeabilized using 0.1% Triton-X-100 (Sigma-Aldrich) in PBS for 5 min. Cells were thoroughly washed with PBS and fixative quenched with 0.1 M glycine for 20 min followed by incubation with 2% BSA fraction V (Thermo, 15260037) in PBS for 30 min. Cells were stained with phalloidin diluted in 2% BSA/PBS at room temperature for 20 min or stained with phalloidin and GFP-booster (ChromoTek, gb2AF488-10) when imaging EGFP-cortactin using 3D-SIM. This was followed by one wash with 1xPBS/0.05% Tween for 10 min and two washes with 1x PBS for 15 min.

## Imaging preparation for live macrophage frustrated phagocytosis

Ham's F-12 (Caisson Labs, UT) supplemented with 2% HI-FBS was used as an imaging medium. Macrophages in 12-wells plates were washed with PBS once and treated with Accutase (ThermoFisher Scientific) at 37 ˚C for 5 min to detach cells. After aspirating Accutase, the cells were gently scraped in the imaging medium and transferred to the IgG-patterned coverslip, which was inserted into an AttoFluor microscope chamber (ThermoFisher A7816). The microscope chamber with the cells was incubated at 37 ˚C for 5 min before transferring to a microscope. For primary mouse macrophages, confocal images were obtained using a Zeiss LSM880 confocal. A plan-Apochromat 63x/1.40 oil objective lens was used. Sequential images were acquired at 5 s intervals (Supplementary Movie 1).

## Structured illumination microscopy (SIM, TIRF-SIM, and 3D-SIM)

SIM and 3D-SIM of fixed macrophages was performed using a Nikon Ti inverted microscope controlled by Nikon NIS-Elements AR software and equipped with a 100x oil immersion objective (1.49 NA, Nikon CFI Apochromat TIRF 100x) and EMCCD camera (Andor DU-897). Three lasers with wavelengths 488, 561, and 647 nm were used. Z-stack image acquisition was controlled by an MCL NanoDrive PiezoZ Drive with a Z step interval 100 nm or 150 nm.

Total internal reflection fluorescence structured illumination microscopy (TIRF-SIM) was performed at the Advanced Imaging Center, HHMI Janelia Research Campus[47]. In brief, the structured illumination pattern was confined to the TIRF-plane of the samples. The TIRF-SIM system was based on a Zeiss Observer.Z1 inverted microscope with an Olympus UApo N 100x oil NA 1.49 objective[48]. Fluorescence emission was recorded using an sCMOS camera (Hamamatsu, Orca Flash 4.0 v2 sCMOS). Three lasers with the wavelengths 488, 560, and 647 nm were used. The time interval between image acquisitions was 5 or 10 s. The samples were mounted in a cell culture chamber (ThermoFisher A7816 AttoFluor chamber) and maintained at 37 °C with 5% $CO_2$.

## Localization-based super-resolution microscopy

To study cortactin with PALM (Supplementary Fig. 8c–g), RAW 264.7 cells were transiently transfected with mEos3.2-cortactin using electroporation as described above and then plated on IgG-patterned coverslips. After fixation with 4% paraformaldehyde, the cells were stained with phalloidin Alexa Fluor 647 (for STORM imaging) and washed with PBS (see Methods section 'Immunofluorescence staining'). Four color fluorescent beads (dilution 1:500, Invitrogen, T7279) were deposited onto the coverslip for drift correction. The cells were sealed in fresh-made STORM buffer[49] (see below) and imaged on a home-built TIRF microscopy immediately. The STORM data was analyzed and rendered with previously published software[50]. Chromatic aberration was corrected using the localization of multi-color beads. Z profiles (Supplementary Fig. 8c–g) were generated by first manually drawing lines (length: $2\,\mu m$, width: $0.5\,\mu m$) across the center of each podosomes (Supplementary Fig. 8c) and Z cross-sections were then extracted.

STORM buffer: Buffer A (10 mM Tris (pH 8.0) + 50 mM NaCl) and Buffer B (50 mM Trix (pH 8.0) + 10 mM NaCl + 10% Glucose) were filtered (Millex-GV, $0.22\,\mu m$, R1BB13647) and stored at RT for further use. GLOX solution $(250\,\mu l)$ consisted of 14 mg Glucose Oxidase (Sigma-Aldrich, G-2133-50ku), $50\,\mu l$ Catalase (Sigma-Aldrich, C100-50MG), and $200\,\mu l$ Buffer A. GLOX solution was stored at 4 °C for up to 2 weeks and spun down at 17,000 g for 3 min before use. The final STORM buffer contained 690 µm Buffer B, $7\,\mu l$ 2-mercaptoethanol (Sigma-Aldrich, M3148-25ML), and $7\,\mu l$ GLOX solution.

## Interferometric photoactivated localization microscopy (iPALM)

For iPALM imaging, RAW 264.7 macrophages were plated on 25 mm diameter round coverslips containing gold nanorod fiducial markers[20,21] used for calibration and drift correction. Histograms were generated for the x, y, and z localizations of 5 gold nanorod fiducials to determine localization precisions. From these histograms, the standard deviation of the FWHM was calculated, resulting in a localization precision of $\sigma_x = 14.68 \pm 5.74$ nm, $\sigma_y = 18.85 \pm 7.85$ nm, $\sigma_z = 5.0 \pm 1.5$ nm (Supplementary Fig. 1b, c). IgG-patterned iPALM coverslips were prepared as described above. First, RAW 264.7 macrophages were plated on the IgG patterned coverslips in Ham's F12 (Caisson Labs, UT) supplemented with 2% HI-FBS medium to allow frustrated phagocytosis for 25 min. Then the cells were fixed with 4% paraformaldehyde at 37 °C for 10–15 min and permeabilized using 0.1% Triton-X-100 (Sigma-Aldrich) in PBS for 5 min. The cells were then thoroughly washed with PBS and quenched with 0.1 M glycine for 20 minutes followed by incubation with 2% BSA fraction V (Thermo, 15260037) in PBS for 30 min. Cells were then stained with phalloidin Alexa Fluor 647 diluted in 2% BSA/PBS at room temperature for 20 min followed by one wash with 1xPBS/0.05% Tween for 10 min and two washes with 1x PBS for 15 min. After fixation and staining, an 18 mm coverslip was added to the top of the sample in STORM buffer[49] and sealed using epoxy. Alexa Fluor 647-labeled samples were excited at 647 nm with 3 kW/cm² laser irradiance at 50 ms exposure time. Fluorescence was collected through a pair of 60×1.49 NA Apo TIRF objectives (Nikon) and 2 emission filters (LP02-647RU and FF01-720/SP (Semrock)). Interference images were digitized via 3 iXon3-DU897E EMCCD cameras (Andor Technologies). Single molecule images, consisting of 400,000 frames, were localized, processed, and rendered using PeakSelector software (Janelia Research Campus)[20,21]. To render iPALM images, we zoomed into areas containing individual frustrated phagocytosis sites and rendered them using Z-stack images with a Z interval of 10 nm, covering the Z range 0–550 nm. To render iPALM images of individual podosomes, we further zoomed into areas containing individual podosomes and rendered them as Z-stack images with a Z interval 5 or 10 nm.

In iPALM images, IgG disks were visible away from the cell due to nonspecific labeling by phalloidin Alexa Fluor 647. The molecule count per µm² was normalized so that the background value was 1.0. This resulted in a normalized molecule count within the cell of 524.4, and within the IgG disks of 5.6 (Supplementary Fig. 4a). Although the IgG value was two orders of magnitude lower, the IgG disk was clearly visible. The molecule count was then assessed at each height to examine the size and localization of the IgG disks (Supplementary Fig. 4b).

## Actin filament analysis

3D volumetric views of iPALM Z-stack images were obtained using a plugin "3D Viewer" in Fiji[51]. iPALM Z-stacks were analyzed using the Imaris 9.5 software package (http://www.bitplane.com). The "FilamentTracer" module was used to detect, trace, visualize and measure radial filaments[52]. Filaments were detected with semi-manual "Autopath tracing", which required manually chosen estimates for the end points and diameter of each filament.

## Initial analysis of phagocytosis molecule distributions

For individual phagocytosis sites, radial averaging was performed for IgG, FcγRIIA, and F-actin ($n = 24$ phagocytosis sites each). Radial distributions were generated using the Radial Profile plugin in ImageJ. For individual IgG and FcγR profiles, a sigmoid function was fit to the profile, and the inflection point of the sigmoid fit was quantified. For the individual F-actin profiles, the peak location was measured.

The same method was used to generate radial distributions and quantify peak locations for myosin II and F-actin ($n = 14$ sites each).

## Automatic identification of podosomes and phagocytosis sites

Analysis was performed in Python (v 3.7.10) using Jupyter Notebook (v 6.1.4) and the package Scipy (v 1.7.3). For a single actin channel, a uniform smoothing filter was applied based on podosome feature size (typically 0.3 μm, input by the user), and a small amount of noise (Gaussian noise with mean = 0, variance = 0.01) was added for pixel uniqueness (Fig. 3a, b, Supplementary Fig. 2a). Persistent homology was performed on these images using the Dionysus2 (v 2.0.6) Python package[53]. A level-set filtration[24,53,54] is generated to describe the topological features within the smoothed images. These filtrations contain a series of simplicial complexes (vertices, edges, and triangular faces), ordered in a descending manner by their associated values (superlevel-set, Supplementary Fig. 2e). The filtrations are created so that each vertex corresponds to a pixel with a value equal to its intensity. Edges are drawn between nearest neighbor vertices (orthogonal) as well as upper-left and lower-right neighboring vertices (diagonal); triangular faces are drawn within three connecting edges (Freudenthal triangulation[53]). Each edge or face is given a value equal to the lowest corresponding vertex (upper-star filtration[24,53,54]). These upper-star, superlevel-set filtrations were then used to generate persistence diagrams (Fig. 3d, e, Supplementary Fig. 2d, e).

Persistence diagrams provided the pixel birth level ($b_{i,j}$) and pixel death level ($d_{i,j}$) for features for a homology group $H_j$. For our analysis on images of frustrated phagocytosis, we use the first two homology groups, where $H_0$ corresponds to connected components and $H_1$ corresponds to holes. The value $b_{i,j}$ is the intensity level, or threshold, at which the $ith$ feature appears for homology $j$. The value $d_{i,j}$ is when a feature merges with an older feature (known as the "Elder rule"). Finally, the persistence values can be defined as $p_{i,j} = |b_{i,j} - d_{i,j}|$. For each homology group ($j = 0,1$), k-means clustering ($n_{clusters} = 2$) was performed on all $p_{i,j}$ to separate significantly persistent, and thus long-lasting, features from short-lived ones. The $b_{i,j}$ of significantly persistent features correspond to the maxima ($j = 0$) or minima ($j = 1$) of this feature, and provide an estimate for the locations of podosomes and phagocytosis sites (Fig. 3f, g, Supplementary Fig. 2b). This is possible because $b_{i,j}$ can be matched to a single pixel due to pixel uniqueness. Note that when analyzing an entire cell (with multiple phagocytosis sites), we generated two separate persistence diagrams per image (Fig. 3), once as above but only for $j = 0$, and a second time using a uniform smoothing filter based on phagocytosis site feature size (typically 1.5 μm, input by user) and only for $j = 1$ (Fig. 3b, c). This step was performed since further blurring the image to look for hole features gave more robust results for finding phagocytosis sites on entire cells, due to the complicated topology of actin between sites (Fig. 3). Additional resources for using level-set filtrations to determine critical points on 2D grids (i.e., images) are documented within the Python packages Dionysus2[53] and Ripser[54].

While the persistent homology step successfully found important features in our data, we needed to include additional refinement steps to (a) locate the centers of the circle of podosomes, (b) only keep podosomes associated with well-defined phagocytosis sites and (c) minimize the false discovery rate (# false positives / # total positives) for podosomes. In essence, podosomes that are far away from sites are dropped, sites with fewer than 3 podosomes are dropped, and a site center is determined by finding the center point of the podosomes associated with a single site (Fig. 3h, Supplementary Fig. 2c).

To validate the pipeline, podosomes were manually selected in four cells with the empirical criterion that they were (a) associated with a well-defined phagocytosis site and (b) had at least 3 podosomes per site. If a podosome was closely identified by both manual detection and the pipeline (to within 0.4 μm), this was labeled a true positive. A false negative was a podosome identified manually but not by the pipeline, and a false positive was a podosome identified by the pipeline but not manually. The false discovery rate was recorded. This process was performed both pre-refinement, and post-refinement.

## iPALM podosome analysis

Podosome and site locations were determined using the automatic identification pipeline from the mean Z-projection for each stack of iPALM actin data ($m = 7$ images/sites). For each podosome, a perpendicular line scan (perpendicular to the circle of podosomes, Fig. 4c) was performed at each height (from Z = 0 to Z = 450 nm, typical Z-step of 10 nm), which consisted of a 1 μm line scan from a podosome toward the center of the phagocytosis site, and a 1 μm line scan away from the center. The radial distribution (Fig. 4d, from r = 0 to r = 600 nm) was calculated for each podosome at each height Z. Individual features, including podosome height and protrusion length, were quantified from each individual radial averaging heatmap (Fig. 5, Supplementary Fig. 3). The contour was generated for each heatmap by smoothing (Gaussian, mean = 0, variance = 1) the heatmap then using a threshold value of the mean intensity within R = 350 nm. Protrusion length was quantified as the difference between the lower section of the lower actin network (the location of the lower part of the FWHM) and the bottom of the podosome (actin contour). Data were interpolated to provide the best quantifications, but results were rounded to the nearest 10 nm to be accurate in terms of our resolution. A 3D contour visualization was generated by rotating the mean radial averaging heatmap 180 degrees around the Z-axis and using the Python package Mayavi (v 4.7.2)[55].

## Analyzing adhesion and cap proteins in relation to podosomes and phagocytosis sites

For paxillin ($n_{cells} = 2$) and talin ($n_{cells} = 4$) TIRF images, podosome and site locations were determined using the automatic identification pipeline using the corresponding actin channels. A perpendicular line scan (1 μm in, 1 μm out) was performed for each species and their corresponding actin channels (Fig. 7c). For each species, the distance from a podosome to the inward and outward peaks, the distance between peaks, and the full-width half max (FWHM) for the peaks were quantified for individual podosomes (Fig. 7d–f).

## Analyzing proteins in relation to podosomes and phagocytosis sites from 3D-SIM data

For 3D-SIM microscopy on experiments with disks of IgG, podosome and site locations were determined using the automatic identification pipeline from the mean Z-projection for each stack of actin for paxillin ($n_{cells} = 4$), α-actinin ($n_{cells} = 4$), myosin ($n_{cells} = 4$), and cortactin ($n_{cells} = 4$). For experiments on uniform IgG (paxillin $n_{cells} = 6$, α-actinin $n_{cells} = 3$) the podosome sites were found using the pipeline, but site locations and refinement were not performed, since phagocytosis sites were not constrained to the disk geometry. For the disk experiments, a perpendicular line scan (1 μm in, 1 μm out for paxillin and α-actinin; 2 μm in, 2 μm out for myosin) was performed at each height Z (varied) for every podosome, and a mean heatmap for single cells was generated (Supplementary Figs. 6, 7, 8b, 9a). For the uniform experiments, radial averaging (max r = 1 μm) was performed at each height Z (varied) for every podosome. From these scans, a mean heatmap for single cells was generated and mirrored for visualization comparisons (Supplementary Figs. 6b, 7b).

## Statistics and reproducibility

Statistics were performed using the Python package Scipy. Significance was determined using a two-tailed Mann-Whitney $U$ test with Bonferroni correction. Line plots were displayed with their 95% confidence interval. Box-and-whisker plots included a box showing the median with first and third quartiles. Whiskers show the last datum

within 1.5*IQR (interquartile range) of the box. Usually, outliers were included as individual points beyond the whiskers.

The number of samples used for quantitative data analysis is indicated in the figure legends. Representative images in figures were obtained from the following number of images and independent experiments: Fig. 1a: 17 images (3 experiments); Fig. 1b: 22 (3), F-actin was labeled with Phalloidin Alexa 568 or Phalloidin Alexa 488; Fig. 1c: 43 (5), F-actin was marked with Lifeact-Halo 549 or FTractin-tdTomato; Fig. 1d: 7 (2); Fig. 2a, c, Fig. 6e and Supplementary Fig. 5d, e: 17 (3); Fig. 6a–d: 20 (3); Fig. 8d, e and Supplementary Fig. 9b: 9 (2); Supplementary Fig. 8a: 13 (2); Supplementary Fig. 8c: 16 (2); Supplementary Fig. 8d–f: 31 (2).

### Reporting summary
Further information on research design is available in the Nature Research Reporting Summary linked to this article.

## Data availability
Most of the data generated in this study have been deposited in Zenodo [https://doi.org/10.5281/zenodo.6657586]. Some of the raw image files were too large to host but these are available upon request from the corresponding authors. All other relevant data supporting the key findings of this study are available within the article and its Supplementary Information files or from the corresponding author upon reasonable request. Source data are provided with this paper.

## Code availability
The code used for this project is available on GitHub [https://github.com/elstonlab/PodosomeImageAnalysis] and on Zenodo [https://doi.org/10.5281/zenodo.6657535][56].

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

## Acknowledgements

We thank Tony Perdue from the Department of Biology Microscopy Core at UNC for assistance with Nikon N-SIM microscopy. The Imaris workstation was provided by the Neuroscience Center Microscopy Core Facility, supported in part by funding from the NIH-NINDS Neuroscience Center Support Grant P30 NS045892 and the NIH-NICHD Intellectual and Developmental Disabilities Research Center Support Grant U54 HD079124. We thank Wolfgang Bergmeier and Juan Song of the University of North Carolina for providing mouse tissue. iPALM and TIRF-SIM imaging were conducted in collaboration with the Advanced Imaging Center at Janelia Research Campus, a facility jointly supported by the Gordon and Betty Moore Foundation and the Howard Hughes Medical Institute. We are grateful to Satya Khuon for cell culture help, Richard Superfine, Michael Falvo, and Timothy O'Brien for help with micropatterning, and Ellen C. O'Shaughnessy for macrophage culture. This work was supported by grants from the National Institute of General Medical Sciences (NIGMS) to KMH (R35 GM122596) and to TCE (R35 GM127145), as well as from the National Institute of Biomedical Imaging and Bioengineering to TCE (U01 EB018816). JCH received support from the NIGMS (5T32 GM067553), MP from the National Institutes of Health (T32 GM008570-21A1), and MEK from the National Science Foundation (NSF-MCB 2005341).

## Author contributions

S.H., T.W., A.T.N., and B.L. performed the experiments and imaging. J.A., A.T., and T.L.C. guided and assisted imaging at Janelia. J.C.H., S.H., T.W., M.P., B.L., M.E.K., and J.A. performed the analyses. J.C.H., S.H., T.C.E., and K.M.H. wrote the manuscript with contributions from all authors. K.M.H. and T.L.C. initiated the study, and the work was directed by T.L.C., T.C.E., and K.M.H.

## Competing interests

The authors declare no competing interests.
