## [Peer Review File · Nature Communications]

REVIEWER COMMENTS

Reviewer #1 (Remarks to the Author):

The manuscript by Herron et al. present the results of a study exploiting advanced imaging and dedicated image analysis to unravel the architecture of phagocytic podosomes at the nanoscale. Although part of the results is a confirmation of previous reports and mostly descriptive, this study has clearly the merit of using iPALM to obtain for the first time a highly detailed axial view of phagocytic podosomes and as such it will be appealing to the NATCOMMS readership. Combining a clever image analysis tool to specifically identify phagocytic podosomes and the iPALM super-resolution capability led the authors to the identification of a “hourglass” shape of the phagocytic podosome actine core. The central finding of this study however needs to be better substantiated before the manuscript can be accepted for publication.

The first main concern I have is the possible influence of the IgG disk topology on the podosome formation. By allowing macrophages to adhere onto a coverslip microprinted with IgG disks and imaging by TIRF-SIM or iPALM, the authors provide a number of quantifications. Considering the IgG disks are a key methodology throughout the paper onto which many calculations are based, I think the authors should provide a nanoscale characterization of these IgG disks. How thick are the disks? Is it possible they are 50-100 nanometer high cylinders and not simply a flat monolayer of IgG molecules? Previous reports have demonstrated that podosomes are preferentially formed at concave or convex edges when cells adhere onto topological surfaces. If the IgG disks used in this study provide some kind of topology, the hourglass shaped actine core observed could in reality be two podosome cores formed at the concave and convex edges of the IgG “cylindrical” disks. Considering the highly regular (vertical) distance between the putative intense upper core and the less intense lower core (i.e. the narrow hourglass neck at 190 ± 10 nm), one should rule out this regularity is due to the IgG disks stamped and not to a real actin feature. I believe that super-resolution images showing both IgG disk fluorescence and phalloidin should be used to show the vertical location of the upper/lower core with respect to the IgG disk fluorescent signal. The authors state this hourglass shape could be specific for phagocytic podosomes and not be present in podosomes formed on homogeneous coating. Was the hourglass shape observed in podosomes far from the IgG disks?

The second major criticism I have is that two sets of data in this manuscript remain a bit hanging in the air. First of all, Figure 1: these data and quantification about the location of FcR signal with respect to the actin signal hang a bit in the air as they are described but nothing further in the remaining manuscript link back to the observation that IgG and FcγR largely tapered away before reaching the podosomes. Are there integrins located at the rim of the IgG disk in between the FcγR and the actin signals? Second example is Supplementary fig 6, which does not go beyond anecdotal observations: some proteins are shown but nothing is done further with these data. Also the text in the discussion (294-298) is highly speculative. This manuscript would stand easily without suppl fig 6.

Minor concern: The observation regarding paxillin having a much denser distribution toward the inside of the phagocytosis site (suppl fig 4) is contrasted to talin but talin by 3D-SIM is not shown.

Reviewer #2 (Remarks to the Author):

The manuscript by Herron et al (Nano-architecture of phagocytic podosomes) describes the structure and organization of phagocytic podosomes using super-resolution techniques. The manuscript is very well written and the images absolutely beautiful. The authors use state-of-the-art imaging techniques and image analysis, adding a strong value to the work. The novel hourglass structure of the F-actin core of the podosome is truly fascinating and definitely significant to the field.

I have some comments:

1) The authors use iPALM for F-actin staining but not for the other components. Therefore the title is slightly misleading, the novel nano-architecture is resolved for F-actin. I would like to see at least some other components placed also into this model, making the manuscript stronger and much more valuable.

2) In the introduction, The authors are referring to the 3D-SIM super-resolution method for studying the podosome structure and cite for according work. However, there is no mentioning about the work done using STORM/dSTORM. This should be included in the introduction section including the its resolution compared to the iPALM.

3) Figure 2c - only few Z-positions are shown making it difficult to see what is described in the text. I would like to see a gallery of the Z-stack (supplemental file). Also, it is not clear, how the actin intensity was measured. It says that the extent of actin outside the podosome was greatly reduced. Was it the actin core that was considered as podosome? Were the radiating actin filaments considered as podosome? What was the distinction between actin in the podosomes and actin outside? This should be more clearly described.

4) The authors did not analyze the podosome core component cortactin. It would be useful to see how the 3D structure of cortactin is, specially in the view of the discovered neck structure. If it is technically feasible, the analysis of F-actin and cortactin simultaneously would be great.

5) I would like to see a model in the end of the article where the hourglass structure with the knob is presented and the other components would be set in the model. Specially I would like to see cortactin here. I understand that the ring components were not analyzed with iPALM and their location is dynamic. But indicative location would be useful. The same applies to the myosin II.

Reviewer #3 (Remarks to the Author):

The manuscript by Herron et al. combining a series of super resolution imaging technologies including the interferometric photoactivated localization microscopy (iPALM), 3D-SIM and TIRF-SIM to study the nano structure of phagocytic podosomes. They also developed a semi-automatic analysis pipeline for identification of podosomes and phagocytosis sites. A novel “hourglass” shape of the podosome core was revealed. The spatial relationship between adhesion, cap proteins and actin were observed.

The paper is well written and is a great example of applying cutting edge imaging techniques to gain insights into biological questions.

Below I present a few comments:

1. It is worthwhile for the author to show the localization precision in iPALM using gold nanorods/nanoparticles fiducial marks in both lateral and axial direction.
2. It is not clear in the paper why certain imaging modality was selected for individual experiments. Therefore, it is important add a few sentences to compare the limits and advantages of iPALM, SIM and TIRF-SIM used in the paper.

Thanks to the reviewers for their valuable comments, which certainly helped us improve the manuscript. We believe we have been able to address all their suggestions and concerns. Thanks also for the supportive comments regarding the work.

REVIEWER COMMENTS

Reviewer #1 (Remarks to the Author):

The manuscript by Herron et al. present the results of a study exploiting advanced imaging and dedicated image analysis to unravel the architecture of phagocytic podosomes at the nanoscale. Although part of the results is a confirmation of previous reports and mostly descriptive, this study has clearly the merit of using iPALM to obtain for the first time a highly detailed axial view of phagocytic podosomes and as such it will be appealing to the NATCOMMS readership. Combining a clever image analysis tool to specifically identify phagocytic podosomes and the iPALM super-resolution capability led the authors to the identification of a “hourglass” shape of the phagocytic podosome actine core. The central finding of this study however needs to be better substantiated before the manuscript can be accepted for publication.

The first main concern I have is the possible influence of the IgG disk topology on the podosome formation. By allowing macrophages to adhere onto a coverslip microprinted with IgG disks and imaging by TIRF-SIM or iPALM, the authors provide a number of quantifications. Considering the IgG disks are a key methodology throughout the paper onto which many calculations are based, I think the authors should provide a nanoscale characterization of these IgG disks. How thick are the disks? Is it possible they are 50-100 nanometer high cylinders and not simply a flat monolayer of IgG molecules? Previous reports have demonstrated that podosomes are preferentially formed at concave or convex edges when cells adhere onto topological surfaces. If the IgG disks used in this study provide some kind of topology, the hourglass shaped actine core observed could in reality be two podosome cores formed at the concave and convex edges of the IgG “cylindrical” disks. Considering the highly regular (vertical) distance between the putative intense upper core and the less intense lower core (i.e. the narrow hourglass neck at 190 ± 10 nm), one should rule out this regularity is due to the IgG disks stamped and not to a real actin feature. I believe that super-resolution images showing both IgG disk fluorescence and phalloidin should be used to show the vertical location of the upper/lower core with respect to the IgG disk fluorescent signal. The authors state this hourglass shape could be specific for phagocytic podosomes and not be present in podosomes formed on homogeneous coating. Was the hourglass shape observed in podosomes far from the IgG disks?

We thank the reviewer for pointing out how our observations might be based on an artefact. We carried out the experiments suggested by the reviewer and are happy to report that the hourglass shape of the podosome was not due to interactions with the IgG disk. The IgG disks' topology could be visualized by nonspecific staining with phalloidin-Alexa. This revealed them to be much closer to the coverslip than even the bottom disk of the podosomes, and well below the podosome neck (IgG disks had a FWHM height of 30 nm, n=11). There were not many examples of podosomes far away from IgG disks, because the IgG disks were arrayed relatively close together, with ~4 μm between disk edges. However, we did observe podosomes that formed over the IgG disk, rather than at the edge of the disks. These podosomes (n = 10) showed the same hourglass shape as those we studied at the edge of the IgG disks. To include this information in the paper, we added a new supplementary figure (Supp. Fig. 4) and text in the Results and Methods sections (pages 8, 23, and 24).

The second major criticism I have is that two sets of data in this manuscript remain a bit hanging in the air. First of all, Figure 1: these data and quantification about the location of FcR signal with respect to the actin signal hang a bit in the air as they are described but nothing further in the remaining manuscript link back to the observation that IgG and FcgR largely tapered away before reaching the podosomes. Are there integrins located at the rim of the IgG disk in between the FcgR and the actin signals?

We include the observations shown in figure 1 to demonstrate that the phagocytosis we see in our model system, induced by IgG circles, resembles that observed using more common models (uniform IgG and fibronectin). FcR and IgG are included because their colocalization indicates that the process is associated with receptor clustering over the IgG, as is normal. This is now mentioned in the figure legend.

This paper focuses largely on the structure of the podosome. We would like to pursue the role of integrin and the dynamics of podosome formation in a following study.

Second example is Supplementary fig 6, which does not go beyond anecdotal observations: some proteins are shown but nothing is done further with these data. Also the text in the discussion (294-298) is highly speculative. This manuscript would stand easily without suppl fig 6.

Per the reviewer's recommendation, we removed Supplementary Fig. 6. The data and discussion regarding the tropomyosin isoforms was removed. Microtubules are now mentioned only in the discussion, with a call out to two supplementary movies. The MT supplementary figure has been removed. We mention the MT data to support a point we make in the discussion, that there may be bridges of actin between podosomes. For α -actinin, only the 3D-SIM data is included now.

Minor concern: The observation regarding paxillin having a much denser distribution toward the inside of the phagocytosis site (suppl fig 4) is contrasted to talin but talin by 3D-SIM is not shown.

We apologize for our unclear writing. Both talin and paxillin were observed by TIRF-SIM, and statistical comparisons of paxillin and talin distributions were from the TIRF-SIM images (Fig. 7). We did observe paxillin using 3D-SIM (Supplementary Fig. 6) as well and noted that, qualitatively, the distribution of paxillin resembled that seen with TIRF-SIM. We have rewritten the text to clarify this (pages 9 & 10).

Reviewer #2 (Remarks to the Author):

The manuscript by Herron et al (Nano-architecture of phagocytic podosomes) describes the structure and organization of phagocytic podosomes using super-resolution techniques. The manuscript is very well written and the images absolutely beautiful. The authors use state-of-the-art imaging techniques and image analysis, adding a strong value to the work. The novel hourglass structure of the F-actin core of the podosome is truly fascinating and definitely significant to the field.

I have some comments:

1) The authors use iPALM for F-actin staining but not for the other components. Therefore the title is slightly misleading, the novel nano-architecture is resolved for F-actin. I would like to see at least some other components placed also into this model, making the manuscript stronger and much more valuable.

We changed the title to “Actin nano-architecture of phagocytic podosomes”. For the revised manuscript we imaged cortactin using 3D PALM/STORM and 3D-SIM, but were unable to add any more iPALM studies because we had no access to the instrument.

2) In the introduction, The authors are referring to the 3D-SIM super-resolution method for studying the podosome structure and cite for according work. However, there is no mentioning about the work done using STORM/dSTORM. This should be included in the introduction section including the its resolution compared to the iPALM.

Work on the podosome structure using STORM/dSTORM is now in the introduction, and we have added comments regarding the resolution and advantages/disadvantages of various superresolution techniques to the discussion. We have also now included some experiments examining cortactin using PALM/STORM (Supplementary Fig. 7).

3) Figure 2c - only few Z-positions are shown making it difficult to see what is described in the text. I would like to see a gallery of the Z-stack (supplemental file). Also, it is not clear, how the actin intensity was measured. It says that the extent of actin outside the podosome was greatly reduced. Was it the actin core

that was considered as podosome? Were the radiating actin filaments considered as podosome? What was the distinction between actin in the podosomes and actin outside? This should be more clearly described.

As requested by the reviewer, we added a figure containing the full Z-stack (Supplemental Fig. 1). Supplementary Movie 3 includes this information as well.

We also added a section describing how we defined inside versus outside the podosome -- when actin was considered to be part of the podosome rather than part of surrounding fibers (pages 5, 6). In figure 2, we first make general observations about the structure and height of the podosome, and later provide precise definitions of what we consider part of the actin core versus external fibers (in discussion of Fig. 5). This section included measurements of actin core and fiber features, leading us to define precisely what was meant by each (see pages 7,8). Based on width, the FWHM of the upper and lower core were typically around 160 and 140 nm respectively, and never passed 300 nm. Thus, anything past this could easily be regarded as actin filaments/networks.

4) The authors did not analyze the podosome core component cortactin. It would be useful to see how the 3D structure of cortactin is, specially in the view of the discovered neck structure. If it is technically feasible, the analysis of F-actin and cortactin simultaneously would be great.

Cortactin and F-actin were imaged using both 3D-SIM and PALM/STORM and a new supplementary figure was added (Supplementary Fig. 7). While 3D-SIM was unable to resolve the hourglass structure, it did show colocalization of actin and cortactin. PALM/STORM did show a 2 lobed actin structure and cortactin present in both lobes.

5) I would like to see a model in the end of the article where the hourglass structure with the knob is presented and the other components would be set in the model. Specially I would like to see cortactin here. I understand that the ring components were not analyzed with iPALM and their location is dynamic. But indicative location would be useful. The same applies to the myosin II.

We tried different approaches to a cartoon showing an overall model. Initially we included information from the literature to generate a somewhat speculative model, but felt we were making too many unfounded assumptions. In the end we decided to illustrate the features that had been observed in our studies. We were able to produce a cartoon showing, to scale, the actin features we had described alongside observations from TIRF-SIM (Fig. 9). Our studies using 3D-SIM did not provide sufficient height resolution relative to iPALM to include 3D information about molecules studied only with 3D-SIM.

Reviewer #3 (Remarks to the Author):

The manuscript by Herron et al. combining a series of super resolution imaging technologies including the interferometric photoactivated localization microscopy (iPALM), 3D-SIM and TIRF-SIM to study the nano structure of phagocytic podosomes. They also developed a semi-automatic analysis pipeline for identification of podosomes and phagocytosis sites. A novel “hourglass” shape of the podosome core was revealed. The spatial relationship between adhesion, cap proteins and actin were observed. The paper is well written and is a great example of applying cutting edge imaging techniques to gain insights into biological questions. Below I present a few comments:

1. It is worthwhile for the author to show the localization precision in iPALM using gold nanorods/nanoparticles fiducial marks in both lateral and axial direction.

We have added this information. Histograms were generated for the x, y, and z localizations of 5 gold nanorod fiducials to determine localization precisions. From these histograms, the standard deviation of the FWHM was calculated, resulting in a localization precision of $\sigma_x = 14.68 \pm 5.74$ nm, $\sigma_y = 18.85 \pm 7.85$ nm, $\sigma_z = 5.0 \pm 1.5$ nm. We also included a 3D view of molecule positions from one representative gold nanorod. This information is now in the methods (page 22) and in Supplementary Fig. 1.

2. It is not clear in the paper why certain imaging modality was selected for individual experiments. Therefore, it is important add a few sentences to compare the limits and advantages of iPALM, SIM and TIRF-SIM used in the paper.

We elaborated on the choices made for imaging, and how advantages and disadvantages affected our selection of different techniques (first paragraph of discussion). We also discuss STORM because it was used in this new version of the manuscript.

REVIEWERS' COMMENTS

Reviewer #1 (Remarks to the Author):

The authors have addressed my concerns and presented convincing new experimental data. I am fully supporting publication of the revised manuscript.

Reviewer #2 (Remarks to the Author):

The revised version of the manuscript Actin nano-architecture of phagocytic podosomes by Herron et al addressed all my concerns. I have no further comments. I suggest the acceptance of the manuscript.

Reviewer #3 (Remarks to the Author):

All my concerns have been addressed. I recommend to publish the manuscript in Nature Communication.